hydrology

lattice Boltzmann method, D1Q3 lattice arrangement, open channel flows, Saint-Venant equations

**Author for correspondence:**
Fengpeng Bai
e-mail: bfp@whu.edu.cn

# A lattice Boltzmann model for the open channel flows described by the Saint-Venant equations

## Zhonghua Yang[1], Fengpeng Bai[2] and Ke Xiang[1]

[1]State Key Laboratory of Water Resources and Hydropower Engineering Science, Wuhan University, Wuhan 430072, People's Republic of China
[2]Changjiang Water Resources Protection Institute, Wuhan 430051, People's Republic of China

FB, 0000-0002-1437-7090

A new lattice Boltzmann method to simulate open channel flows with complex geometry described by a conservative form of Saint-Venant equations is developed. The Saint-Venant equations include an original treatment of the momentum equation source term. Concrete hydrostatic pressure thrust expressions are provided for rectangular, trapezoidal and irregular cross-section shapes. A D1Q3 lattice arrangement is adopted. External forces, such as bed friction and the static term, are discretized with a centred scheme. Bounce back and imposed boundary conditions are considered. To verify the proposed model, four cases are carried out: tidal flow over a regular bed in a rectangular cross-section, steady flow in a channel with horizontal and vertical contractions, steady flow over a bump in a trapezoidal channel and steady flow in a non-prismatic channel with friction. Results indicate that the proposed scheme is simple and can provide accurate predictions for open channel flows.

## 1. Introduction

Numerical modelling of one-dimensional (1D) open channel flows described by a conservative form of Saint-Venant equations [1–4] is a central topic in hydraulic and hydrologic research. Conventional computational schemes mainly focus on the discretization of partial differential equations. For instance, Moussa & Bocquillon [5] investigated the parameter ranges of the finite difference method. Liang & Marche [6] proposed a well-balanced numerical scheme to simulate shallow frictional flows involving wetting and drying by using a Godunov-type scheme. Chang *et al.* [7] presented a mesh-less numerical model based on smoothed particle hydrodynamics to simulate 1D open channel flows. Murillo & García-Navarro [8] solved Saint-Venant equations by applying the energy balanced property.

The lattice Boltzmann (LB) method is a relatively new discrete numerical approach that has elicited increasing attention recently. The method is characterized by simple calculation, parallel process and easy implementation of boundary conditions, and is very efficient and flexible to simulate different flows within complex/varying geometries. It is these features that make the LB method a very promising computational method in different areas. In the area of simulating the open channel flows described by Saint-Venant equations, the LB method is suitable for subcritical flows, which are the most common scenarios in coastal areas, estuaries and rivers. It suffers from a numerical instability when the LB method is used to solve the supercritical flows. The LB method involves streaming and collision steps. The advantages of the LB method, such as simplicity, efficiency and easy treatment of boundary conditions, in simulating fluid flows have been demonstrated [9,10]. Unlike conventional numerical methods, the LB method describes macroscopic fluid flows from the microscopic flow behaviour through particle distribution functions. The LB method was first derived based on the lattice gas automata [11]. The Bhatnagar–Gross–Krook (BGK) scheme has made the LB method simple and efficient [12]. Salmon [13] and Zhou [10] developed LB method theories for modelling shallow water flows. Mayer *et al.* [14] carried out the simulations of a subchannel of a rod bundle with triangular rod arrangement using the LB method. Rasin *et al.* [15] and Peng *et al.* [16] solved the advection–diffusion equation with a multi-relaxation lattice kinetic method. Fernandino *et al.* [17] proposed an LB method in conjunction with the Smagorinsky subgrid scale (SGS) model to simulate the turbulent open duct flow. Van Thang *et al.* [18] discussed the accuracy and stability of the LB method on a D1Q3 lattice and applied the method to a canal network with various hydraulic interconnection structures. Considering the vegetation elements as solid boundaries in flows, Gac [19] presented a 3D lattice model and computed the vertical velocity profile in an open channel flow. Liu *et al.* [20] proposed an LB model to solve the 1D non-conservative form of Saint-Venant equations under the assumption that the width change of river cross-sections is inconspicuous along the stream-wise direction.

An LB model with a D1Q3 lattice arrangement was developed in this study to solve a conservative form of Saint-Venant equations (LBCSVE). Compared with the former LB models to solve the Saint-Venant equations [19,20], the LBCSVE applied the conservative form of Saint-Venant equations for the first time and the Gauss–Legendre numerical integration method was used to solve the hydrostatic pressure thrust in the LB model first. The model was verified in four cases: tidal flow over a regular bed in a rectangular cross-section, steady flow in a channel with horizontal and vertical contractions, steady flow over a bump in a trapezoidal channel and steady flow in a non-prismatic channel with friction.

The rest of this paper is organized as follows. Section 2 presents the governing equations and computing methods of the hydrostatic pressure thrust term for regular and irregular cross-section shapes, and the constructed LB model. Section 3 presents an evaluation of the scheme's performance in four cases. Section 4 provides the conclusions.

## 2. Material and methods

### 2.1. Governing equations

The conservative form of 1D Saint-Venant equations describes shallow water flows in natural rivers and channels. That is

$$\frac{\partial \mathbf{u}}{\partial t} + \frac{\partial \mathbf{f}}{\partial x} = \mathbf{s}, \tag{2.1}$$

$$\mathbf{u} = \begin{bmatrix} A \\ Q \end{bmatrix}; \quad \mathbf{f} = \begin{bmatrix} Q \\ \dfrac{Q^2}{A} + gI_1 \end{bmatrix}; \quad \mathbf{s} = \begin{bmatrix} 0 \\ -gAS_f + g\dfrac{\partial I_1}{\partial x}|_{\bar{z}} \end{bmatrix} \tag{2.2}$$

and

$$I_1 = \int_0^{h_z} (h_z - h_i)b(x, h_i)\,\mathrm{d}h_i, \tag{2.3}$$

where $t$ is time; $x$ is the stream-wise coordinate; $A$ is the wetted cross-sectional area; $Q$ is discharge; $g$ is gravitational acceleration; $\bar{z}$ is a constant water level [21]; $b(x, h_i)$ is the channel width on the water surface, $z_i = z_b + h_i$ (figure 1); $I_1$ is the hydrostatic pressure thrust resulting from longitudinal width variation; and $S_f$ is a friction term modelled by Manning's formula and is expressed as

$$S_f = \frac{n^2Q|Q|}{R^{4/3}A^2}; \quad R = \frac{A}{P}, \tag{2.4}$$

where $n$ is the roughness coefficient, $R$ is the hydraulic radius and $P$ is the wetted perimeter.

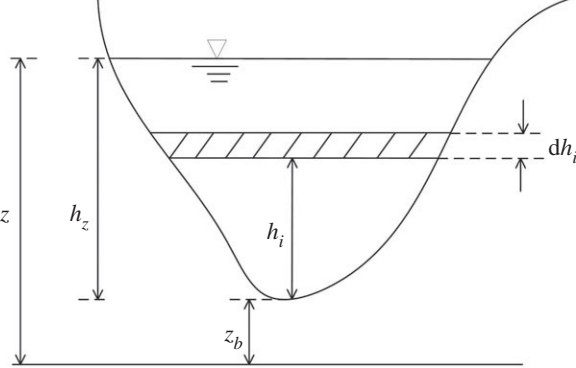

**Figure 1.** Cross-section shape and variable definition.

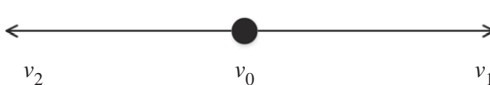

**Figure 2.** D1Q3 lattice arrangement.

**Table 1.** Calculation formulae of the hydrostatic pressure thrust $I_1$.

| cross-section shape | $I_1$ |
| --- | --- |
| rectangular cross-section | $Bh_z^2/2$ |
| trapezoidal cross-section | $dh_z^2/2 + mh_z^3/3$ |
| irregular cross-section | $(h_z/2)\sum_{l=1}^{n_l} C_l f\left((h_z/2)h_{G_l} + h_z/2\right)$ |

## 2.2. Calculation of the integral $I_1$

$I_1$ represents hydrostatic pressure thrust through integration on the wetted cross-section in consideration of the distance of each infinitesimal area element with respect to the water surface. Table 1 shows the calculation formulae of $I_1$ for channels with rectangular and trapezoidal cross-section shapes and natural rivers with irregular cross-section shapes. The Gauss–Legendre numerical integration method is applied for irregular cross-section shapes (see appendix A).

## 2.3. Lattice Boltzmann model

We considered the LB method to solve the conservative form of 1D Saint-Venant equations on a D1Q3 lattice arrangement (shown in figure 2). The discrete velocities [18,20] are

$$v_\alpha = \begin{cases} 0 & \alpha = 0 \\ v & \alpha = 1 \\ -v & \alpha = 2, \end{cases} \tag{2.5}$$

where $\alpha$ is the link in a lattice; $v = \Delta x/\Delta t$ denotes the velocity along a lattice link, with $\Delta x$ being the lattice and $\Delta t$ being the time step. In D1Q3 lattice arrangement, each lattice has two links ($v_1$ and $v_2$) to its neighbours. $v_0$ indicates that the particle stays at its original lattice without movement.

The LB method involves two steps: streaming and collision. In the streaming step, the particles move to the neighbouring lattice points in their directions and at their velocities governed by

$$f_\alpha(x + v_\alpha\Delta t, t + \Delta t) = f_\alpha'(x, t) + w_\alpha\frac{\Delta t}{c_s^2}v_\alpha F, \tag{2.6}$$

where $f_\alpha$ denotes the distribution function of particles, $f_\alpha'$ is the value before the streaming step, $c_s^2$ is a constant and equal to $v^2/3$, $F$ is the external force and $w_\alpha$ represents the weight factor determined by the pattern of the lattice, which is

$$w_0 = \frac{2}{3}; \quad w_1 = w_2 = \frac{1}{6}. \tag{2.7}$$

In the collision step, $f'_\alpha(x, t)$ is expressed as

$$f'_\alpha(x, t) = f_\alpha(x, t) + \Omega_\alpha(f_\alpha(x, t)),\tag{2.8}$$

where $\Omega_\alpha$ is the collision operator. The BGK model is used due to its simplicity and efficiency

$$\Omega_\alpha(f_\alpha) = -\frac{1}{\tau}(f_\alpha - f_\alpha^{eq}),\tag{2.9}$$

where $\tau$ is the single relaxation time and $f_\alpha^{eq}$ denotes the local equilibrium distribution function.

Equations (2.6), (2.8) and (2.9) are combined to obtain the evolution equation with single-relaxation time as follows:

$$f_\alpha(x + v_\alpha \Delta t, t + \Delta t) = f_\alpha(x, t) - \frac{1}{\tau}(f_\alpha - f_\alpha^{eq}) + w_\alpha \frac{\Delta t}{c_s^2} v_\alpha F.\tag{2.10}$$

The local equilibrium distribution plays an essential role in the LB method. It decides what flow equations are to be solved. $f_\alpha^{eq}$ must satisfy the following three conditions, namely, mass, momentum conservation and momentum tensor in equations (2.1) and (2.2)

$$\sum_\alpha f_\alpha^{eq} = A,\tag{2.11}$$

$$\sum_\alpha v_\alpha f_\alpha^{eq} = Q\tag{2.12}$$

and

$$\sum_\alpha v_\alpha^2 f_\alpha^{eq} = \frac{Q^2}{A} + gI_1.\tag{2.13}$$

For the D1Q3 lattice arrangement, the local equilibrium distributions can be expressed as (see appendix B)

$$f_0^{eq} = A - \frac{gI_I}{v^2} - \frac{Q^2}{Av^2},\tag{2.14}$$

$$f_1^{eq} = \frac{gI_I}{2v^2} + \frac{Q}{2v} + \frac{Q^2}{2Av^2}\tag{2.15}$$

and

$$f_2^{eq} = \frac{gI_I}{2v^2} - \frac{Q}{2v} + \frac{Q^2}{2Av^2}.\tag{2.16}$$

The macroscopic variables are defined as

$$A = \sum_\alpha f_\alpha, \quad Q = \sum_\alpha v_\alpha f_\alpha.\tag{2.17}$$

External force $F$ in equation (2.6) is the source term in the momentum equation in equations (2.1)–(2.2) and is expressed as

$$F = -gAS_f + g\frac{\partial I_1}{\partial x}\big|_z.\tag{2.18}$$

The centred scheme proposed by Zhou [10] was applied in this study. The scheme has second order in space and time. The external force term was evaluated at the mid-point between the lattice point and its neighbouring lattice point as

$$F_\alpha = F_\alpha\left(x + \frac{1}{2}v_\alpha \Delta t\right).\tag{2.19}$$

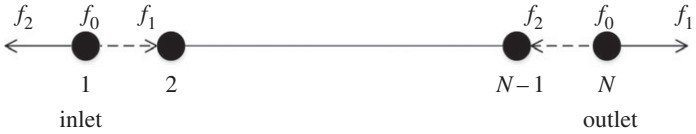

**Figure 3.** Distribution function at the inlet and outlet boundaries.

Substitution of equation (2.18) into equation (2.19) leads to

$$F_\alpha = -gA(x)S_f(x),\tag{2.20}$$

$$F_1 = -g\overline{A_1}\,\overline{S_{f_1}} + g\left(\frac{I_1(x + v\Delta t) - I_1(x)}{\Delta x}\right)_{\bar{z}},\tag{2.21}$$

$$\overline{A_1} = \frac{A(x + v\Delta t) + A(x)}{2}, \quad \overline{S_{f_1}} = \frac{S_f(x + v\Delta t) + S_f(x)}{2},\tag{2.22}$$

$$F_2 = -g\overline{A_2}\,\overline{S_{f_2}} + g\left(\frac{I_1(x) - I_1(x - v\Delta t)}{\Delta x}\right)_{\bar{z}}\tag{2.23}$$

and

$$\overline{A_2} = \frac{A(x - v\Delta t) + A(x)}{2}, \quad \overline{S_{f_2}} = \frac{S_f(x - v\Delta t) + S_f(x)}{2}.\tag{2.24}$$

### 2.3.1. Boundary conditions

As shown in figure 3, $f_2$ and $f_0$, at the inlet boundary can be obtained after the streaming step. Unknown distribution function $f_1$ (shown as a dashed line) cannot be determined from the internal lattice nodes. Also, at the outlet boundary, $f_2$ (shown as a dashed line) is unknown. Proper boundary conditions are necessary to determine the unknown distribution functions.

*Bounce-back boundary condition.* The basic idea of the bounce-back condition is that an incoming particle towards the boundary bounces back into the fluid. At the inlet boundary, incoming unknown distribution function $f_1$ is equal to $f_2$. Similarly, unknown distribution function $f_2$ is equal to $f_1$ at the outlet boundary.

*Imposed boundary condition.* Specific variable values are commonly applied at boundaries. For example, constant discharge $Q_{in}$ and fixed water level $z_{out}$ are imposed at inlet and outlet boundaries, respectively. For the inlet boundary, the treatment involves three steps. In the first step, a zero-gradient condition for water level $z$ is set, and wetted area $A$ (2.1) is calculated

$$z(1) = z(2).\tag{2.25}$$

Second, the velocity is calculated as

$$u(1) = \frac{Q_{in}}{A(1)}.\tag{2.26}$$

Third, the distribution function $f_1$ is calculated as

$$f_1 = f_1^{eq} + f_2 - f_2^{eq}.\tag{2.27}$$

$f_1^{eq}$ is computed based on the macroscopic variables obtained in the first and second steps.

Unknown distribution function $f_2$ at the outlet boundary can be calculated through the same steps. First, a zero-gradient condition for discharge $Q$ is set

$$Q(N) = Q(N - 1).\tag{2.28}$$

Second, wetted area $A(N)$ is calculated with fixed water level $z_{out}$. The velocity is

$$u(N) = \frac{Q(N)}{A(N)}.\tag{2.29}$$

Third, $f_2$ is determined as

$$f_2 = f_2^{eq} + f_1 - f_1^{eq}.\tag{2.30}$$

### 2.3.2. Stability conditions

The magnitude of the resultant velocity is smaller than velocity $v$ along a lattice link and celerity [10]

$$\frac{u^2}{v^2} < 1\tag{2.31}$$

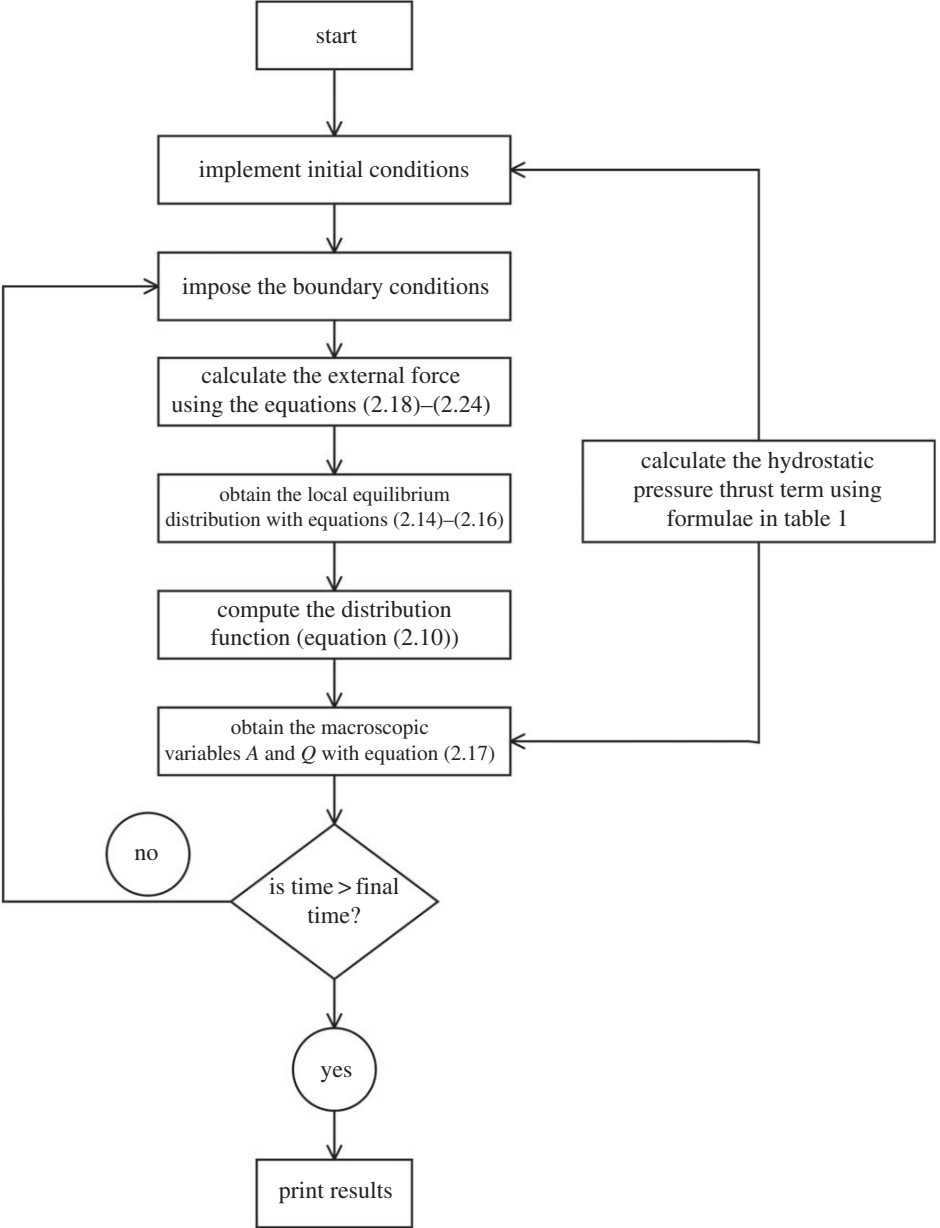

**Figure 4.** Flowchart showing the basic steps of the LBCSVE model.

and

$$\frac{gh}{v^2} < 1. \tag{2.32}$$

The essential procedures of the LBCSVE model are summarized in figure 4.

## 3. Numerical tests

The LBCSVE model was validated through four benchmark tests.

### 3.1. Tidal flow over a regular bed

We considered the test proposed by Bermudez & Vazquez [22] used to verify an upwind discretization of bed slope source terms. Bed elevation is defined as (figure 5a)

$$z_b(x) = 10 + \frac{40x}{L} + 10\sin\left[\pi\left(\frac{4x}{L} - 0.5\right)\right], \tag{3.1}$$

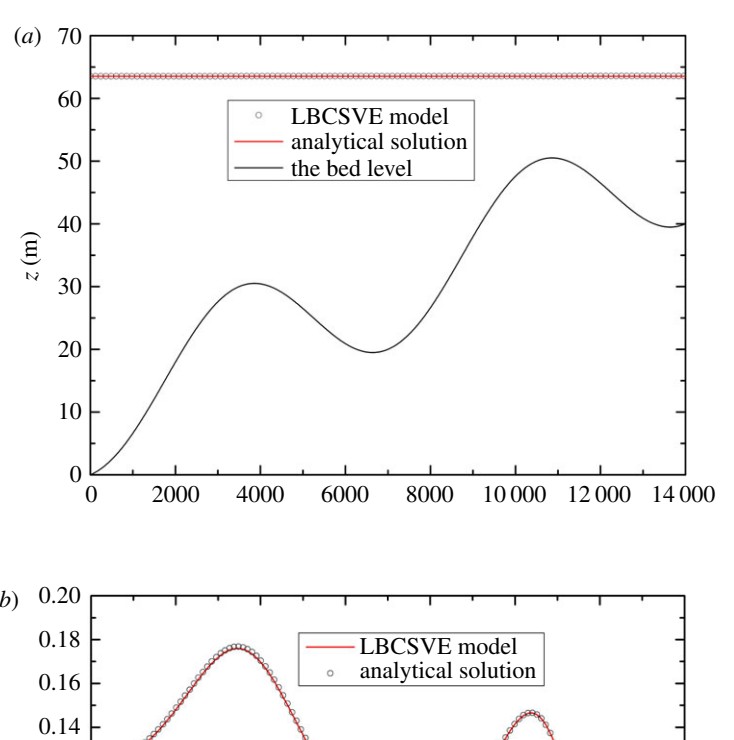

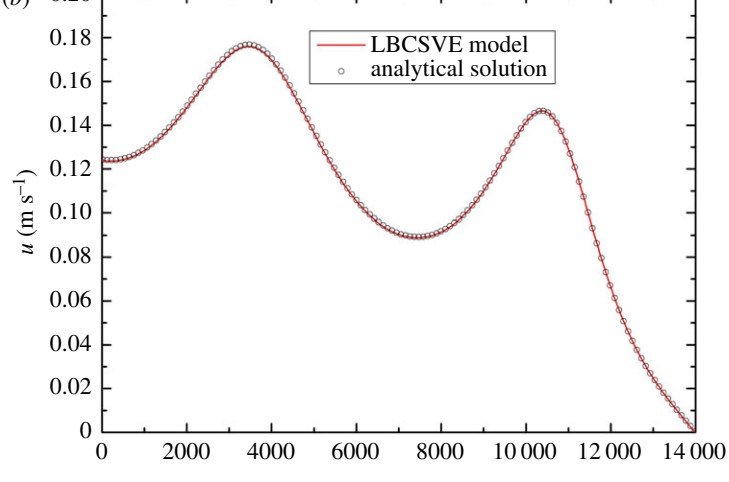

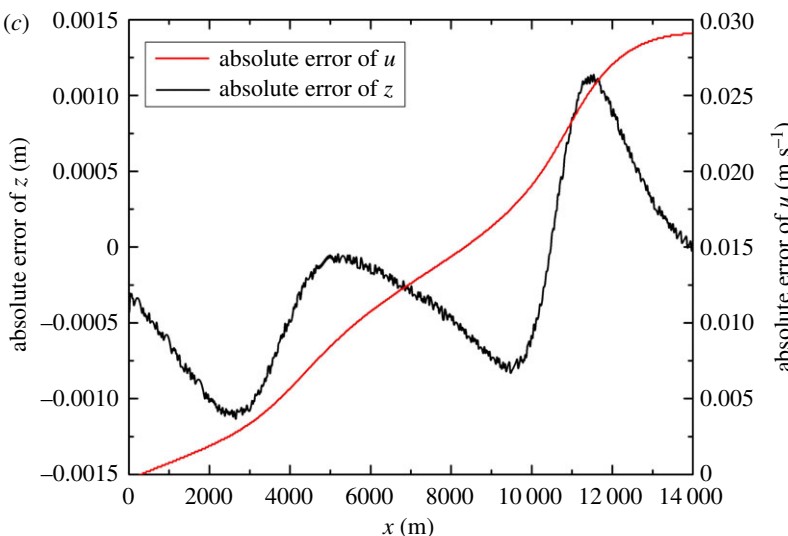

**Figure 5.** Tidal wave flow in an open channel at $t = 9$, 117.5 s and comparison of the analytical solution and numerical results. (*a*) Water level, (*b*) velocity and (*c*) the absolute errors of water level and velocity.

where $L = 14\,000$ m is the channel length. The initial condition is

$$\left.\begin{array}{l} z(x, 0) = 60.5 \text{ m} \\ Q(x, 0) = 0.0\,\text{m}^3\,\text{s}^{-1}, \end{array}\right\} \tag{3.2}$$

and

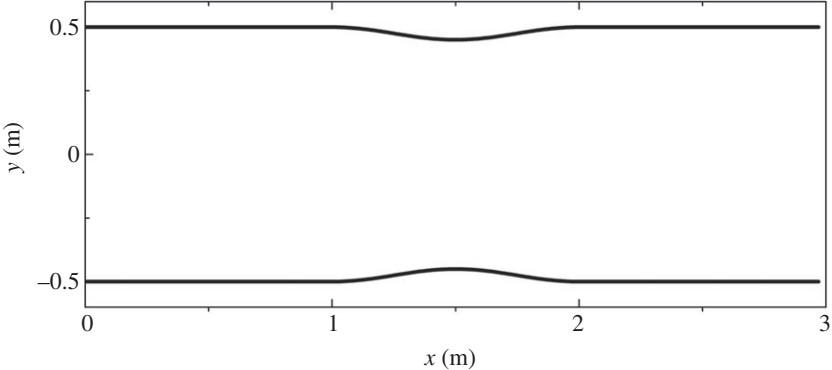

**Figure 6.** Channel with vertical contractions in §3.2: top view.

($a$)

**Figure 7.** Steady flow in a channel with horizontal and vertical contractions and comparison of the analytical solution and numerical results: ($a$) water level and ($b$) discharge.

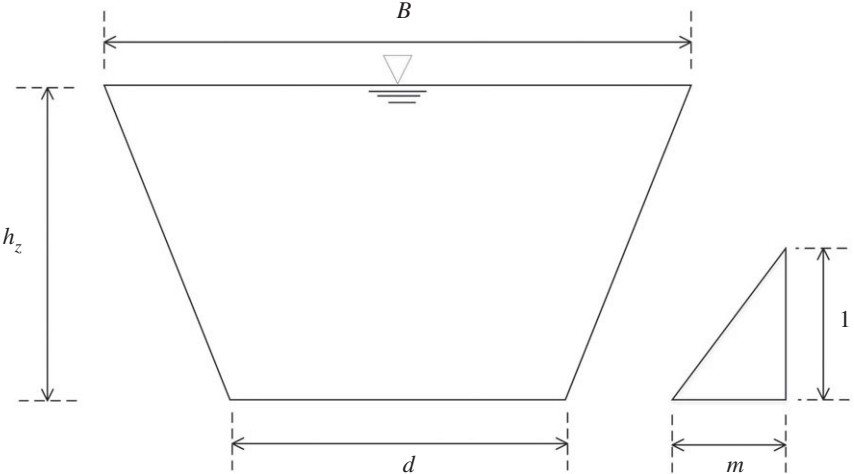

**Figure 8.** Trapezoidal cross-section and variable definition.

and the inlet and outlet boundary conditions are

$$z(0, t) = 64.5 - 4.0 \sin\left[\pi\left(\frac{4t}{86{,}400} + 0.5\right)\right]$$
and
$$Q(L, t) = 0.0. \qquad (3.3)$$

In the computations, the rectangular and frictionless channel width is 1.0 m, with $\Delta x = 17.8$ m and $\Delta t = 0.3$ s. The relaxation time $\tau = 0.6$. Figure 5 presents a comparison of the numerical results and the asymptotic analytical solution at $t = 9, 117.5$ s. Good agreements were observed.

## 3.2. Steady flow in a channel with horizontal and vertical contractions

For this case [23], the channel is frictionless and 3 m long. Simulation was undertaken to reproduce steady flows with varying breadth and topography. Channel breadth (shown in figure 6) and topography are provided by

$$z_b(x) = \begin{cases} 0.1 \cos^2\left[\pi(x - 1.5)\right] & \text{if } |x - 1.5| < 0.5 \\ 0 & \text{otherwise} \end{cases} \qquad (3.4)$$

and

$$b(x) = \begin{cases} 1 - 0.1 \cos^2[\pi(x - 1.5)] & \text{if } |x - 1.5| < 0.5 \\ 1 & \text{otherwise.} \end{cases} \qquad (3.5)$$

A unit discharge of $q = 1.566$ m$^2$ s$^{-1}$ was imposed at the inflow, and a depth of 1 m was fixed for the outflow. In the computation, the number of lattice nodes was 100, and the lattice speed was $v = 6$ m s$^{-1}$. The relaxation time $\tau = 0.9$. A steady-state solution was obtained (shown in figure 7). The predicted surface profile in figure 7a matches the analytical one perfectly. The value of discharge predicted with the proposed LBCSVE model is equal to exactly 1.566 m$^2$ s$^{-1}$. Figure 7b shows that the LBCSVE model offers a better solution than the model proposed by Alias *et al.* [23] based on the finite-volume Godunov-type framework with a slight oscillatory behaviour near the bump.

## 3.3. Steady flow over a bump in a trapezoidal channel

The steady flow over a bump is a classical test problem used as a benchmark test case for numerical methods by many researchers [24–26]. The channel is 25 m long, and topography is defined as

$$z_b = \begin{cases} 0.2 - 0.05(x - 10)^2 & \text{if } 8 < x < 12 \\ 0 & \text{otherwise.} \end{cases} \qquad (3.6)$$

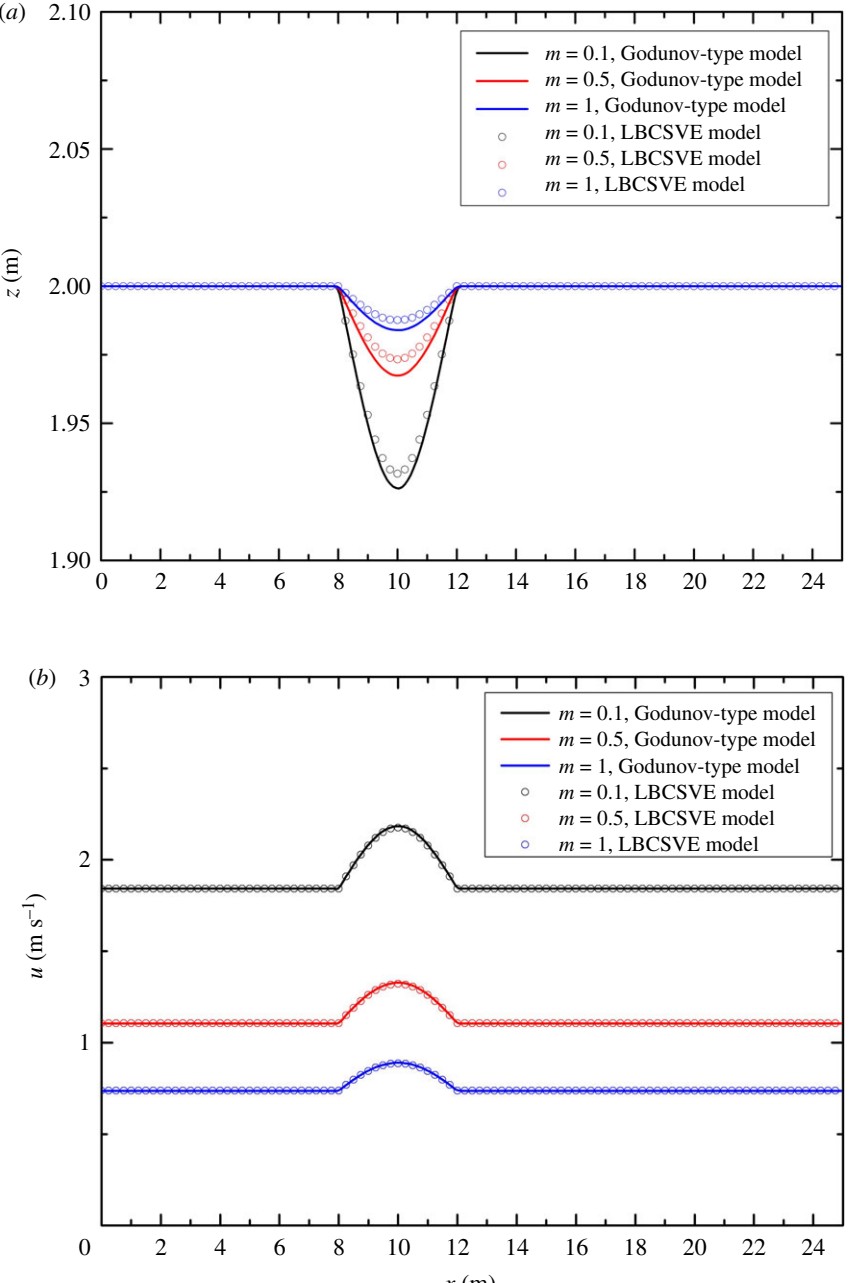

**Figure 9.** Steady flow over a bump in a trapezoidal channel and comparison of the LBCSVE model and the Godunov-type model: (*a*) water level and (*b*) velocity.

The trapezoidal cross-section (shown in figure 8) was adopted in this test case. The channel width $d(x)$ was 1.0 m. The relaxation time $\tau = 1.0$. Several values of the slope coefficient ($m = 0.1, 0.5, 1$) were selected. The cell size of $\Delta x = 0.1$ m was used for both LBCSVE and Godunov-type models with $v = 10$ m s$^{-1}$. A comparison of the results of the finite-volume Godunov-type framework with Harten, Lax and van Leer approximate Riemann solvers [6,27,28] is plotted in figure 9. Good agreement was observed. The quantitative comparison indicates that the maximum relative error for water level is smaller than 0.34% for the three slope coefficients.

## 3.4. Steady flow in a non-prismatic channel with friction

This test case was developed by MacDonald [29]. The analytical solution of steady flows in a non-prismatic channel with friction exists when channel width and water depth are given. The Manning coefficient, $n$, is 0.03 m s$^{-1/3}$. The channel length is 400 m with a trapezoidal cross-section. The slope

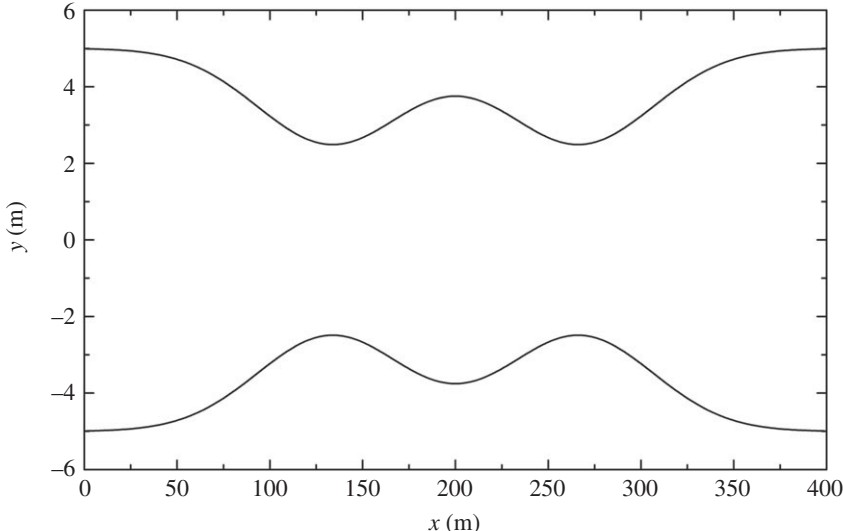

**Figure 10.** Steady flow in a non-prismatic channel with friction in §3.4: top view.

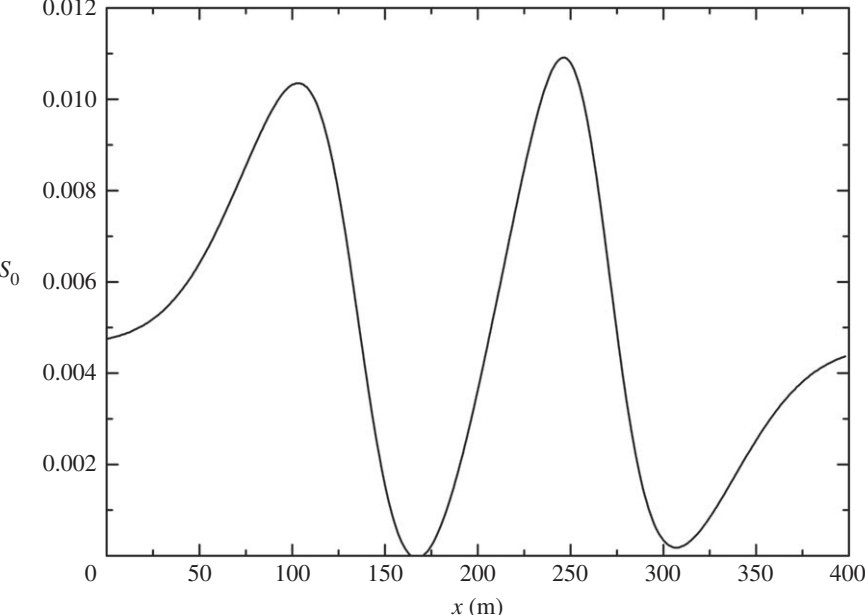

**Figure 11.** Steady flow in a non-prismatic channel with friction: bed slope.

coefficient, $m$, is equal to 2. At the inlet, the discharge is $20 \, \text{m}^3 \, \text{s}^{-1}$, and a fixed downstream depth of 0.904094 m is provided. The channel width is (shown in figure 10)

$$b(x) = 10 - 5\exp\left(-50\left(\frac{x}{400} - \frac{1}{3}\right)^2\right) - 5\exp\left(-50\left(\frac{x}{400} - \frac{2}{3}\right)^2\right). \tag{3.7}$$

Figure 11 shows the bed slope defined by bed width $b$, Manning coefficient $n$, slope coefficient $m$, discharge $Q$ and water depth $h$

$$S_0 = \left(1 - \frac{Q^2(b + 2mh)}{gh^3(b + mh)^3}\right)\frac{\partial h}{\partial x} + Q^2 n^2 \frac{(b + 2h\sqrt{1 + m^2})^{4/3}}{h^{10/3}(b + mh)^{10/3}} - \frac{Q^2(\partial b/\partial x)}{gh^2(b + mh)^3}, \tag{3.8}$$

where grid size $\Delta x$ is 2 m and time step $\Delta t$ is 0.1 s. The relaxation time $\tau = 0.6$. Figure 12 proves that the LBCSVE model can predict the water level and velocity accurately.

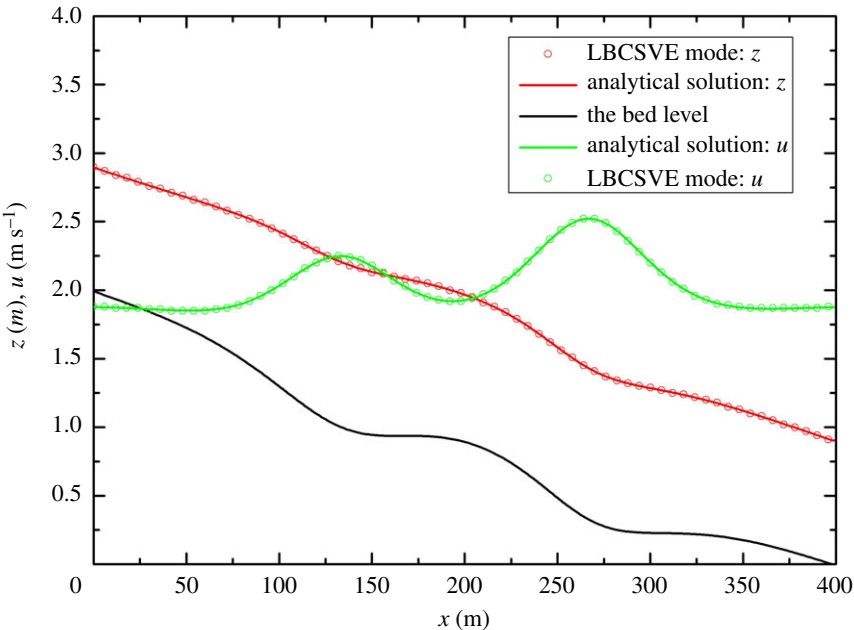

**Figure 12.** Steady flow in a non-prismatic channel with friction and comparison of the analytical solution and the numerical results: water level and velocity.

## 4. Conclusion

In this study, a new LB model with a D1Q3 lattice (LBCSVE) was developed to solve a conservative form of Saint-Venant equations. The LBCSVE model can provide accurate predictions for 1D frictional open channel flows with various cross-section shapes.

The general mathematical formulation proposed in this paper encourages the application to other case studies.

Data accessibility. The data and Fortran source code have been uploaded to the Dryad Digital Repository: https://dx.doi.org/10.5061/dryad.dj5g9v3 [30].

Authors' contributions. F.B. conceived the research; Z.Y. performed the research; F.B. and K.X. contributed to model development; Z.Y. and F.B. interpreted the results. All authors gave final approval for publication.

Competing interests. The authors declare no conflict of interest.

Funding. F.B. was funded by the National Key Research and Development Program of China (grant no. 2016YFC0402207); Z.Y. and K.X. was funded by the National Natural Science Foundation of China (grant nos 51679170, 51879199 and 51439007).

## Appendix A

No exact expression of $I_1$ is available for natural rivers with irregular cross-section shapes. The Gauss–Legendre numerical integration method is thus presented. The Legendre polynomial $L_n(x)$ is expressed as

$$L_{n_l}(x) = \frac{1}{2^{n_l} n!} \frac{\mathrm{d}^{n_l}}{\mathrm{d}x^{n_l}} [(x^2 - 1)^{n_l}]. \tag{A 1}$$

The Gauss–Legendre integration formula is presented as

$$\int_{-1}^{1} f(x)\mathrm{d}x \approx \sum_{l=1}^{n_l} C_l f(x_l) \tag{A 2}$$

and

$$C_l = \int_{-1}^{1} \frac{L_{n_l}(x)}{(x - x_l)L'_{n_l}(x)} \, \mathrm{d}x \quad (l = 1, 2 \cdots n_l), \tag{A 3}$$

where $n_l$ is the number of integration points and $C_l$ is the integration coefficient.

For $I_1$, we define

$$f(h_i) = (h_z - h_i)b(x, h_i) \quad \text{(A 4)}$$

and

$$h_i = \frac{h_z}{2}(h_G + 1). \quad \text{(A 5)}$$

By applying equations (A 4) and (A 5), $I_1$ becomes

$$I_1 = \int_0^{h_z} (h_z - h_i)b(x, h_i)\mathrm{d}h_i$$
$$= \frac{h_z}{2}\int_{-1}^{1} f\left(\frac{h_z}{2}h_G + \frac{h_z}{2}\right)\mathrm{d}h_G \quad \text{(A 6)}$$
$$\approx \frac{h_z}{2}\sum_{l=1}^{n_l} C_l f\left(\frac{h_z}{2}h_{G_l} + \frac{h_z}{2}\right).$$

# Appendix B

The local equilibrium distribution plays an essential role in the LB method. It decides what flow equations are to be solved. For the D1Q3 lattice arrangement, $f_a^{\mathrm{eq}}$ is assumed to be a polynomial [31], that is

$$f_\alpha^{\mathrm{eq}} = A_\alpha + B_\alpha v_\alpha u + C_\alpha u^2. \quad \text{(B 1)}$$

Where $A_\alpha$, $B_\alpha$ and $C_\alpha$ are the coefficients to calculate and $u$ is macroscopic velocity equal to $Q/A$. Given that the local equilibrium distribution has the same symmetry as the lattice (figure 2), we have

$$A_1 = A_2 = \bar{A}$$
$$B_1 = B_2 = \bar{B}$$
$$C_1 = C_2 = \bar{C}. \quad \text{(B 2)}$$

$f_\alpha^{\mathrm{eq}}$ must satisfy the following three conditions, namely, mass, momentum conservation and momentum tensor in equations (2.1) and (2.2)

$$\sum_\alpha f_\alpha^{\mathrm{eq}} = A, \quad \text{(B 3)}$$

$$\sum_\alpha v_\alpha f_\alpha^{\mathrm{eq}} = Q \quad \text{(B 4)}$$

and

$$\sum_\alpha v_\alpha^2 f_\alpha^{\mathrm{eq}} = \frac{Q^2}{A} + gI_1. \quad \text{(B 5)}$$

Substitution of equation (B 1) into equations (B 3)–(B 5) results in

$$A_0 + 2\bar{A} + (C_0 + 2\bar{C})u^2 = A, \quad \text{(B 6)}$$
$$2\bar{B}v^2 u = Q \quad \text{(B 7)}$$

and

$$2v^2\bar{A} + 2\bar{C}v^2 u^2 = \frac{Q^2}{A} + gI_I. \quad \text{(B 8)}$$

From equation (B 7), we obtain

$$B_1 = B_2 = \bar{B} = \frac{Q}{2v^2 u}. \quad \text{(B 9)}$$

After evaluating the terms in equations (B 6) and (B 7) and equating the coefficients of $A$ and $u^2$, respectively, we have

$$A_0 + 2\bar{A} = A, \quad \text{(B 10)}$$
$$C_0 + 2\bar{C} = 0, \quad \text{(B 11)}$$
$$2v^2\bar{A} = gI_I \quad \text{(B 12)}$$

and

$$2\bar{C}v^2 u^2 = \frac{Q^2}{A}. \quad \text{(B 13)}$$

The solutions of equations (B 10)–(B 13) result in

$$A_0 = A - \frac{gI_I}{v^2}, \tag{B 14}$$

$$C_0 = -\frac{Q^2}{Av^2u^2}, \tag{B 15}$$

$$A_1 = A_2 = \bar{A} = \frac{gI_I}{2v^2} \tag{B 16}$$

and

$$C_1 = C_2 = \bar{C} = \frac{Q^2}{2Av^2u^2}. \tag{B 17}$$

Substitutions of equations (B 9) and (B 14)–(B 17) into equation (B 1) lead to the local equilibrium distribution

$$f_0^{\mathrm{eq}} = A - \frac{gI_I}{v^2} - \frac{Q^2}{Av^2}, \tag{B 18}$$

$$f_1^{\mathrm{eq}} = \frac{gI_I}{2v^2} + \frac{Q}{2v} + \frac{Q^2}{2Av^2} \tag{B 19}$$

and

$$f_2^{\mathrm{eq}} = \frac{gI_I}{2v^2} - \frac{Q}{2v} + \frac{Q^2}{2Av^2}. \tag{B 20}$$

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
