## [Reviewer comments · Royal Society Open Science]

Review History

RSOS-190439.R0 (Original submission)

Review form: Reviewer 1 (Jingming Hou)

Is the manuscript scientifically sound in its present form?

Yes

Are the interpretations and conclusions justified by the results?

Yes

Is the language acceptable?

Yes

Is it clear how to access all supporting data?

Not Applicable

Do you have any ethical concerns with this paper?

No

Have you any concerns about statistical analyses in this paper?

No

Recommendation?

Accept with minor revision (please list in comments)

Comments to the Author(s)

Authors describe the method of simulation 1D open-channel flow by means of one of simplest LBM algorithms with the D1Q3 lattice arrangement. As a base, the system of Saint-Venant equations is used. The Saint-Venant equations are formulated in a conservative form with inclusion of two pressure terms. The idea to solve the conservative form of the Saint-Venant equations by using the LBM is by my opinion a step forward in attempt to develop robust 1D LBM solver for practical application.

Manuscript is in general correct and should be interesting for readers but it requires some minor corrections before publication:

1. The similar model has made appearance in <Study of the 1D lattice Boltzmann shallow water equation and its coupling to build a canal network. > Thang and Chopard et. al., J. of Comput. Phys., 229 (2010) 7373-7400. The authors should explain the difference compared with it.
2. In the second section, the governing equations had better be structured in non-dimensional form, since the original equation the LB method is trying to solve is in its dimensionless form.
3. The Chapman-Enskog expansion should be included to show exactly which governing equations can be recovered, if a brand new LB model is developed. However, considering the simplicity of the paper, the Chapman-Enskog expansion has better show to the reviewers and may not appeared in the paper.
4. Detailed description of obtaining of equilibrium distribution function in Sec. 3 may interrupt the train of thought of readers. I would recomend move everything between eq. (24) and (42) to an appendix.
5. Fig. 4 should be greater to improve its readability.
6. Some plots also require for improve their readability, e.g. Fig. 6 maight contain only one half of a channel while it is symmetric and then the vertical axis could have the sacle from, let say, 0.4 to 0.6; Fig. 7b is also nearly empty, vertical scale 1.56-1.57 would improve it.

After these corrections I recommend the paper for publication.

Review form: Reviewer 2**Is the manuscript scientifically sound in its present form?**

No

Are the interpretations and conclusions justified by the results?

No

Is the language acceptable?

Yes

Is it clear how to access all supporting data?

Yes

Do you have any ethical concerns with this paper?

No

Have you any concerns about statistical analyses in this paper?

I do not feel qualified to assess the statistics

Recommendation?

Reject

Comments to the Author(s)

The manuscript proposed the LBCSVE scheme for open channels flow. However, the paper is not clear about its originality and not convincing by presenting four cases of steady flow. Based on this, the reviewer do not recommend this manuscript to be published in current form.

Major concerns:

The originality of the LBCSVE should be discussed in the introduction. Why is this a new lattice Boltzmann method as stated in the abstract?

The advantage and disadvantage of the lattice Boltzmann method should be discussed.

The numerical test only include steady state solution. Does this imply the method only apply to steady state? If not, unsteady flow should be simulated.

In the numerical tests, numerical accuracy and errors should be reported, computational efficiency compared to other numerical methods (eg. Finite difference method, finite volume method) should be provided. The effects of lattice size should also be investigated.

Minor points:

Figure 9, there is no need to provide area A , since it's related to surface elevation, results of flowrate should be given instead.

Figure 12, results of flowrate should be given.

There are some typos in the manuscript, for example m^2/s , superscript should be used.

Review form: Reviewer 3

Is the manuscript scientifically sound in its present form?

Yes

Are the interpretations and conclusions justified by the results?

Yes

Is the language acceptable?

Yes

Is it clear how to access all supporting data?

Yes

Do you have any ethical concerns with this paper?

No

Have you any concerns about statistical analyses in this paper?

No

Recommendation?

Major revision is needed (please make suggestions in comments)

Comments to the Author(s)

The manuscript presents a lattice Boltzmann model for the Saint-Venant equations.

This topic is of potential interest but the actual contribution in the present form is still improvable. The English of the paper is good with a coherent structure. To my opinion the structure of the paper is the correct as the classic Introduction/methods/results and discussion/conclusion is followed.

The novelty of the paper is not well described and presented. To the reader (including myself) it is hard to see what is effectively the novelty of the paper since all the methodologies, as far as I could understand, have already been presented in other research. Images are of very low quality and results not perceptible in the manuscript images.

To my opinion there are some flaws in the paper that have to be clarified regarding the novelty of the paper and as such, I cannot recommend that this paper is accepted for publication in its present form. As such my advice is to review.

Comments:

- Abstracts should not have acronyms
- P4 EQ5 - define alpha
- P5 L20 - the subscript should be alpha
- Eq 5 to Eq 24 and the whole methodology is similar to [18] (Van Thang, P.; Chopard, B.; Lefèvre, L. Study of the 1D lattice Boltzmann shallow water equation and its coupling to build a canal network. J. Comput. Phys. 2010, 229, 7373-7400. (doi: 10.1016/j.jcp.2010.06.022)). The only novelty I found relevant is the use of the variable A instead of h. Please state very explicitly what is the real novelty of this paper.
- P6 L11 - Zhou's scheme for the force does not conserve mass locally. You should clearly state, or at least give an insight on why you have almost no errors since the scheme you used is not conservative.
- P8 L53 - Please compute an absolute error - L2-norm or something similar is advised. Also, for each test the computational time and the machine used should be stated so that there is a comparison of efficiency between your methodology and others.
- P9 L35 - Why use $v=6\text{m/s}$? please explain the value of v used for this and other tests
- P11 L25 - $\text{m/s}^{1/3}$. - fix units
- P13 Table A2 - Table is unnecessary
- references are formatted incorrectly. Some have an initial indentation

Decision letter (RSOS-190439.R0)

13-Jun-2019

Dear Dr Bai,

The editors assigned to your paper ("A lattice Boltzmann model for the open channel flows

described by the Saint-Venant equations") have now received comments from reviewers. We would like you to revise your paper in accordance with the referee and Associate Editor suggestions which can be found below (not including confidential reports to the Editor). Please note this decision does not guarantee eventual acceptance.

Please submit a copy of your revised paper before 06-Jul-2019. Please note that the revision deadline will expire at 00.00am on this date. If we do not hear from you within this time then it will be assumed that the paper has been withdrawn. In exceptional circumstances, extensions may be possible if agreed with the Editorial Office in advance. We do not allow multiple rounds of revision so we urge you to make every effort to fully address all of the comments at this stage. If deemed necessary by the Editors, your manuscript will be sent back to one or more of the original reviewers for assessment. If the original reviewers are not available, we may invite new reviewers.

- Data accessibility

If you wish to submit your supporting data or code to Dryad (<http://datadryad.org/>), or modify your current submission to dryad, please use the following link:
<http://datadryad.org/submit?journalID=RSOS&manu=RSOS-190439>

- Competing interests

- Authors' contributions

- Acknowledgements

- Funding statement

on behalf of Dr Mark Smith (Associate Editor) and Jon Blundy (Subject Editor)
openscience@royalsociety.org

Associate Editor's comments (Dr Mark Smith):

My recommendation is that major revision is needed before this manuscript can be accepted for publication. In particular, I note that each of the reviewers has requested greater clarity on the novelty of the method used. I also suggest that these are described in greater detail which may help address the novelty issue.

Each reviewer raises substantial areas for clarity and correction. I suggest these are addressed before the manuscript is returned.

Comments to Author:

Reviewers' Comments to Author:

Reviewer: 1

Comments to the Author(s)

Authors describe the method of simulation 1D open-channel flow by means of one of simplest LBM algorithms with the D1Q3 lattice arrangement. As a base, the system of Saint-Venant equations is used. The Saint-Venant equations are formulated in a conservative form with inclusion of two pressure terms. The idea to solve the conservative form of the Saint-Venant equations by using the LBM is by my opinion a step forward in attempt to develop robust 1D LBM solver for practical application.

Manuscript is in general correct and should be interesting for readers but it requires some minor corrections before publication:

1. The similar model has made appearance in <Study of the 1D lattice Boltzmann shallow water equation and its coupling to build a canal network. > Thang and Chopard et. al., J. of Comput. Phys., 229 (2010) 7373–7400. The authors should explain the difference compared with it.
2. In the second section, the governing equations had better be structured in non-dimensional form, since the original equation the LB method is trying to solve is in its dimensionless form.
3. The Chapman-Enskog expansion should be included to show exactly which governing equations can be recovered, if a brand new LB model is developed. However, considering the simplicity of the paper, the Chapman-Enskog expansion has better show to the reviewers and may not appeared in the paper.
4. Detailed description of obtaining of equilibrium distribution function in Sec. 3 may interrupt the train of thought of readers. I would recomend move everything between eq. (24) and (42) to an appendix.
5. Fig. 4 should be greater to improve its readability.
6. Some plots also require for improve their readability, e.g. Fig. 6 maight contain only one half of a channel while it is symmetric and then the vertical axis could have the sacle from, let say, 0.4 to 0.6; Fig. 7b is also nearly empty, vertical scale 1.56-1.57 would improve it.

After these corrections I recommend the paper for publication.

Reviewer: 2

Comments to the Author(s)

The manuscript proposed the LBCSVE scheme for open channels flow. However, the paper is not clear about its originality and not convincing by presenting four cases of steady flow. Based on this, the reviewer do not recommend this manuscript to be published in current form.

Major concerns:

The originality of the LBCSVE should be discussed in the introduction. Why is this a new lattice Boltzmann method as stated in the abstract?

The advantage and disadvantage of the lattice Boltzmann method should be discussed.

The numerical test only include steady state solution. Does this imply the method only apply to steady state? If not, unsteady flow should be simulated.

In the numerical tests, numerical accuracy and errors should be reported, computational efficiency compared to other numerical methods (eg. Finte difference method, finite volume method)should be provided. The effects of lattice size should also be investigated.

Minor points:

Figure 9, there is no need to provide area A, since it's realted to surface elevation, results of flowrate should be given instead.

Figure 12, results of flowrate should be given.

There are some typos in the manuscript, for example m²/s, superscript should be used.

Reviewer: 3

Comments to the Author(s)

The manuscript presents a lattice Boltzmann model for the Saint-Venant equations.

This topic is of potential interest but the actual contribution in the present form is still improvable. The English of the paper is good with a coherent structure. To my opinion the structure of the paper is the correct as the classic Introduction/methods/results and discussion/conclusion is followed.

The novelty of the paper is not well described and presented. To the reader (including myself) it is hard to see what is effectively the novelty of the paper since all the methodologies, as far as I could understand, have already been presented in other research. Images are of very low quality and results not perceptible in the manuscript images.

To my opinion there are some flaws in the paper that have to be clarified regarding the novelty of the paper and as such, I cannot recommend that this paper is accepted for publication in its present form. As such my advice is to review.

Comments:

- Abstracts should not have acronyms
- P4 EQ5 - define alpha
- P5 L20 - the subscript should be alpha
- Eq 5 to Eq 24 and the whole methodology is similar to [18] (Van Thang, P.; Chopard, B.; Lefèvre, L. Study of the 1D lattice Boltzmann shallow water equation and its coupling to build a canal network. J. Comput. Phys. 2010, 229, 7373-7400. (doi: 10.1016/j.jcp.2010.06.022)). The only novelty I found relevant is the use of the variable A instead of h. Please state very explicitly what is the real novelty of this paper.
- P6 L11 - Zhou's scheme for the force does not conserve mass locally. You should clearly state, or at least give an insight on why you have almost no errors since the scheme you used is not conservative.
- P8 L53 - Please compute an absolute error - L2-norm or something similar is advised. Also, for each test the computational time and the machine used should be stated so that there is a comparison of efficiency between your methodology and others.
- P9 L35 - Why use v=6m/s? please explain the value of v used for this and other tests
- P11 L25 - m/s^{1/3}. - fix units
- P13 Table A2 - Table is unnecessary
- references are formatted incorrectly. Some have an initial indentation

Author's Response to Decision Letter for (RSOS-190439.R0)

See Appendix A.

RSOS-190439.R1 (Revision)

Review form: Reviewer 2

Is the manuscript scientifically sound in its present form?

Yes

Are the interpretations and conclusions justified by the results?

Yes

Is the language acceptable?

Yes

Do you have any ethical concerns with this paper?

No

Have you any concerns about statistical analyses in this paper?

No

Recommendation?

Accept as is

Comments to the Author(s)

The reviewer (former #2) is satisfied with the modification to improve its quality and recommend the manuscript to be accepted.

Review form: Reviewer 3

Is the manuscript scientifically sound in its present form?

Yes

Are the interpretations and conclusions justified by the results?

Yes

Is the language acceptable?

Yes

Do you have any ethical concerns with this paper?

No

Have you any concerns about statistical analyses in this paper?

No

Recommendation?

Accept as is

Comments to the Author(s)

Dear Authors

Thank you for having incorporated the changes and the improvements suggested in the preceding round of the revision process. I do believe that the manuscript was greatly improved in terms of readability and scientific soundness. Overall I found the manuscript to be much easier to understand, with the aims, methodology, results and significance of the work more evident to the reader. The Authors are to be congratulated for the substantial revision of their manuscript.

Decision letter (RSOS-190439.R1)

04-Oct-2019

Dear Dr Bai,

I am pleased to inform you that your manuscript entitled "A lattice Boltzmann model for the open channel flows described by the Saint-Venant equations" is now accepted for publication in Royal Society Open Science.

Kind regards,
Anita Kristiansen
Royal Society Open Science Editorial Office
Royal Society Open Science
openscience@royalsociety.org

on behalf of Dr Mark Smith (Associate Editor) and Jon Blundy (Subject Editor)
openscience@royalsociety.org

Associate Editor Comments to Author (Dr Mark Smith):

Associate Editor:

Comments to the Author:

Thank you for resubmitting "A lattice Boltzmann model for the open channel flows described by the Saint-Venant equations". to Royal Society Open Science. I have received 2 further reviews of your revised manuscript, which are included below and/or attached. As you can see, both reviewers are satisfied that you have addressed all points raised with the original submission and I am happy to recommend that we accept your contribution in its present form.

Reviewer comments to Author:

Reviewer: 2

Comments to the Author(s)

The reviewer (former #2) is satisfied with the modification to improve its quality and recommend the manuscript to be accepted.

Reviewer: 3

Comments to the Author(s)

Dear Authors

Thank you for having incorporated the changes and the improvements suggested in the preceding round of the revision process. I do believe that the manuscript was greatly improved in terms of readability and scientific soundness. Overall I found the manuscript to be much easier to understand, with the aims, methodology, results and significance of the work more evident to the reader. The Authors are to be congratulated for the substantial revision of their manuscript.

Appendix A

Response to Associate Editor:

Comments: My recommendation is that major revision is needed before this manuscript can be accepted for publication. In particular, I note that each of the reviewers has requested greater clarity on the novelty of the method used. I also suggest that these are described in greater detail which may help address the novelty issue.

Each reviewer raises substantial areas for clarity and correction. I suggest these are addressed before the manuscript is returned.

Response: Thanks so much for giving us many opportunities to revise our manuscript. We appreciate the editor and reviewers very much for their constructive comments and suggestions. I have read the reviewer's suggestions and made corresponding revision one by one. We really carefully revise this paper marked with red color and hope to meet with your approval.

Response to Reviewer#1:

Comments to the Author(s)

Authors describe the method of simulation 1D open-channel flow by means of one of simplest LBM algorithms with the D1Q3 lattice arrangement. As a base, the system of Saint-Venant equations is used. The Saint-Venant equations are formulated in a conservative form with inclusion of two pressure terms. The idea to solve the conservative form of the Saint-Venant equations by using the LBM is by my opinion a step forward in attempt to develop robust 1D LBM solver for practical application.

Manuscript is in general correct and should be interesting for readers but it requires some minor corrections before publication:

Reviewer#1, Concern # 1: The similar model has made appearance in <Study of the 1D lattice Boltzmann shallow water equation and its coupling to build a canal network. > Thang and Chopard et. al., J. of Comput. Phy., 229 (2010) 7373–7400. The authors should explain the difference compared with it.

Response: In Thang et. al's work, the governing equations(Saint-Venant equations), are expressed as

$$\frac{\partial h}{\partial t} + \frac{\partial(hu)}{\partial x} = 0$$
$$\frac{\partial(hu)}{\partial t} + \frac{\partial(hu^2 + 0.5gh^2)}{\partial x} = F$$

The model proposed by Thang et. al is applied to simulate a canal network without considering the variation of river width and the cross-section shape. So, the model cannot be used to simulate the real rivers.

In our work, the governing equations are

$$\frac{\partial \mathbf{u}}{\partial t} + \frac{\partial \mathbf{f}}{\partial x} = \mathbf{s}$$

$$\mathbf{u} = \begin{bmatrix} A \\ Q \end{bmatrix}; \quad \mathbf{f} = \begin{bmatrix} Q \\ \frac{Q^2}{A} + gI_1 \end{bmatrix}; \quad \mathbf{s} = \begin{bmatrix} 0 \\ -gAS_f + g \frac{\partial I_1}{\partial x} \Big|_z \end{bmatrix}$$

$$I_1 = \int_0^{h_z} (h_z - h_i) b(x, h_i) dh_i$$

The primitive variables are the wetted cross-section area A and the discharge Q . The Equations are the traditional and conservative forms [1-3] and can be used in real rivers with arbitrary cross-section shapes. In order to calculate the hydrostatic pressure thrust I_1 which was a difficulty in simulating the real rivers, the Gauss–Legendre numerical integration method was used.

Reviewer#1, Concern # 2: In the second section, the governing equations had better be structured in non-dimensional form, since the original equation the LB method is trying to solve is in its dimensionless form.

Response: Thanks for the valuable suggestions. Originally, the LB method is do trying to solve is the dimensionless equation form. Recently, in the area of simulating the free-surface flows using LB method, many researchers have done a lot work and do not use the dimensionless form [4-10]. The form of governing equations in Thang et. al's work is also not dimensionless form. For the LB model for free-surface flows used the dimensional form, two stability conditions must be satisfied:

(1)The relaxation time $\tau > 0.5$

(2)The magnitude of the resultant velocity is smaller than the speed of calculated with the lattice speed:

$$\frac{u^2}{v^2} < 1$$

And also the celerity.

$$\frac{gh}{v^2} < 1$$

which u is the water velocity, h is the water depth, v is the lattice speed.

Reviewer#1, Concern # 3:The Chapman-Enskog expansion should be included to show exactly which governing equations can be recovered, if a brand new LB model

is developed. However, considering the simplicity of the paper, the Chapman-Enskog expansion has better show to the reviewers and may not appeared in the paper.

Response: Thanks for the valuable suggestions. The recovery of shallow water equations which is the hyperbolic conservative form using the Chapman-Enskog expansion method have been carried out in many works^[4,9,11]. Due to the similar Chapman-Enskog analysis, we do not put the Chapman-Enskog expansion in our work. The following is the Chapman-Enskog expansion for our model.

The Saint-Venant equations can be derived from the Chapman-Enskog expansion.

Assuming Δt is small and equal ε .

Eq.(10) in the manuscript can be expressed

$$f_\alpha(x+v_\alpha\varepsilon,t+\varepsilon)-f_\alpha(x,t)=-\frac{1}{\tau}(f_\alpha-f_\alpha^{eq})+w_\alpha\frac{\varepsilon}{c_s^2}v_\alpha F \quad (R1)$$

The Eq. (R1) is taken a Taylor expansion to the left hand side of in the time and space

$$\begin{aligned} \varepsilon\left(\frac{\partial}{\partial t}+v_\alpha\frac{\partial}{\partial x}\right)f_\alpha+\frac{1}{2}\varepsilon^2\left(\frac{\partial}{\partial t}+v_\alpha\frac{\partial}{\partial x}\right)^2f_\alpha+o(\varepsilon^3) \\ =-\frac{1}{\tau}(f_\alpha-f_\alpha^{eq})+w_\alpha\frac{\varepsilon}{c_s^2}v_\alpha F \end{aligned} \quad (R2)$$

Then, expanding the f_α around the f_α^0 gives

$$f_\alpha=f_\alpha^0+\varepsilon f_\alpha^1+o(\varepsilon^2) \quad (R3)$$

In which $f_\alpha^0=f_\alpha^{eq}$. To order ε , Eq. (R2) becomes

$$\left(\frac{\partial}{\partial t}+v_\alpha\frac{\partial}{\partial x}\right)f_\alpha^0=-\frac{1}{\tau}f_\alpha^1+w_\alpha\frac{1}{c_s^2}v_\alpha F \quad (R4)$$

And to order ε^2 is

$$\left(\frac{\partial}{\partial t}+v_\alpha\frac{\partial}{\partial x}\right)f_\alpha^1+\frac{1}{2}\left(\frac{\partial}{\partial t}+v_\alpha\frac{\partial}{\partial x}\right)^2f_\alpha^0=0 \quad (R5)$$

Substitution Eq.(R4) into Eq.(R5) leads to

$$\left(1-\frac{1}{2\tau}\right)\left(\frac{\partial}{\partial t}+v_\alpha\frac{\partial}{\partial x}\right)f_\alpha^1=-\frac{1}{2}\left(\frac{\partial}{\partial t}+v_\alpha\frac{\partial}{\partial x}\right)\left(w_\alpha\frac{1}{c_s^2}v_\alpha F\right) \quad (R6)$$

Taking \sum_{α} (Eq.(R4)+ ε Eq.(R6)) gives

$$\frac{\partial}{\partial t} (\sum_{\alpha} f_{\alpha}^0) + \frac{\partial}{\partial x} (\sum_{\alpha} v_{\alpha} f_{\alpha}^0) = -\frac{1}{2c_s^2} \frac{\partial}{\partial x} (\sum_{\alpha} w_{\alpha} v_{\alpha} F) \quad (\text{R7})$$

Considering the $f_{\alpha}^0 = f_{\alpha}^{eq}$ and evaluating the above equation using Eq.(5), Eq.(7) and Eqs.(14)- (17) in the manuscript gives the continuity equation

$$\frac{\partial A}{\partial t} + \frac{\partial Q}{\partial x} = 0 \quad (\text{R8})$$

Then taking the $\sum_{\alpha} v_{\alpha}$ (Eq.(R4)+ ε Eq.(R6)) produces

$$\begin{aligned} & \frac{\partial}{\partial t} (\sum_{\alpha} v_{\alpha} f_{\alpha}^0) + \frac{\partial}{\partial x} (\sum_{\alpha} v_{\alpha} v_{\alpha} f_{\alpha}^0) + \varepsilon (1 - \frac{1}{2\tau}) \frac{\partial}{\partial x} (\sum_{\alpha} v_{\alpha} v_{\alpha} f_{\alpha}^1) \\ & = -\frac{1}{2c_s^2} \frac{\partial}{\partial x} (\sum_{\alpha} v_{\alpha} (\frac{\partial}{\partial t} + v_{\alpha} \frac{\partial}{\partial x}) (w_{\alpha} v_{\alpha} F)) \end{aligned} \quad (\text{R9})$$

Then, considering the $f_{\alpha}^0 = f_{\alpha}^{eq}$ and evaluating the above equation using Eq.(5), Eq.(7) and Eqs.(14)- (17) in the manuscript again, the momentum equation is obtained

$$\frac{\partial Q}{\partial t} + \frac{\partial (\frac{Q^2}{A} + gI_1)}{\partial x} = F \quad (\text{R10})$$

The kinematic viscosity is

$$\nu = \Delta t c_s^2 (\tau - \frac{1}{2}) \quad (\text{R11})$$

Reviewer#1, Concern # 4: Detailed description of obtaining of equilibrium distribution function in Sec. 3 may interrupt the train of thought of readers. I would recomend move everything between eq. (24) and (42) to an appendix.

Response: Thanks for this valuable suggestion. The detailed description of obtaining of equilibrium distribution function is moved to the Appendix 2.

Reviewer#1, Concern # 5: Fig. 4 should be greater to improve its readability.

Response: Thanks for this valuable suggestion. We have replaced the original Fig. 4 with a higher resolution figure.

Reviewer#1, Concern # 6: Some plots also require for improve their readability, e.g.

Fig. 6 might contain only one half of a channel while it is symmetric and then the vertical axis could have the scale from, let say, 0.4 to 0.6; Fig. 7b is also nearly empty, vertical scale 1.56-1.57 would improve it.

Response: Thanks for this valuable suggestion. For more readable, the vertical scale in Fig. 7b was set from 1.565 to 1.569. For Fig. 6, in order to show the channel characteristic of vertical contractions, we think it is better to display the two sides of the channel.

References:

- [1] Franzini F, Soares-Frazão S. Efficiency and accuracy of Lateralized HLL, HLLS and Augmented Roe's scheme with energy balance for river flows in irregular channels [J]. *Applied Mathematical Modelling*, 2016, 40(17–18):7427-7446.
- [2] Zhang S, Duan J G. 1D finite volume model of unsteady flow over mobile bed[J]. *Journal of Hydrology*, 2011, 405(1–2):57-68.
- [3] Chang T J, Kao H M, Chang K H, et al. Numerical simulation of shallow-water dam break flows in open channels using smoothed particle hydrodynamics[J]. *Journal of Hydrology*, 2011, 408(1):78-90.
- [4] Frandsen J B. Free-surface lattice Boltzmann modeling in single phase flows[M]// *Advanced Numerical Models For Simulating Tsunami Waves And Runup*. 2008:163-219.
- [5] Zhang J, Zhang Q, Qiao G. A lattice Boltzmann model for the non-equilibrium flocculation of cohesive sediments in turbulent flow[J]. *Computers & Mathematics with Applications*, 2014, 67(2):381-392.
- [6] Rocca M L, Montessori A, Prestininzi P, et al. A multispeed Discrete Boltzmann Model for transcritical 2D shallow water flows[J]. *Journal of Computational Physics*, 2015, 284:117-132.
- [7] Liu H, Zhou J G, Burrows R. Lattice Boltzmann simulations of the transient shallow water flows[J]. *Advances in Water Resources*, 2010, 33(4):387-396.
- [8] Xiong W, Zhang J. A two-dimensional lattice Boltzmann model for uniform channel flows[J]. *Computers & Mathematics with Applications*, 2011,

61(12):3453-3460.

- [9] Thang P V, Chopard B, Mendes E, et al. Study of the 1D lattice Boltzmann shallow water equation and its coupling to build a canal network[J]. Journal of Computational Physics, 2010, 229(19):7373-7400.
- [10] Zhou J G, Haygarth P M, Withers P J, et al. Lattice Boltzmann method for the fractional advection-diffusion equation[J]. Physical Review E, 2016, 93(4-1):043310.
- [11] Zhou J G. Lattice Boltzmann Methods for Shallow Water Flows [M]. 2003.

Response to Reviewer#2:

Comments to the Author(s)

The manuscript proposed the LBCSVE scheme for open channels flow. However, the paper is not clear about its originality and not convincing by presenting four cases of steady flow. Based on this, the reviewer do not recommend this manuscript to be published in current form.

Major concerns:

Reviewer#2, Concern # 1: The originality of the LBCSVE should be discussed in the introduction. Why is this a new lattice Boltzmann method as stated in the abstract?

Response: Thanks for this valuable suggestion. In the introduction, page 2, we add the following statement to prove the originality for the proposed LBCSVE:

Compared with the former LB models to solve the Saint–Venant equations [19, 20], the LBCSVE applied the conservative form of Saint–Venant equations for the first time and the Gauss–Legendre numerical integration method was used to solve the hydrostatic pressure thrust in the LB model first.

Reviewer#2, Concern # 2: The advantage and disadvantage of the lattice Boltzmann method should be discussed.

Response: Thanks for this valuable suggestion. We have add the discussions about the advantage and disadvantage of the lattice Boltzmann method in the introduction, which is:

The method is characterized by simple calculation, parallel process, easy implementation of boundary conditions and very efficient, flexible to simulate different flows within complex/varying geometries. It is these features that make the lattice Boltzmann method a very promising computational method in different areas. In the area of simulating the open channel flows described by Saint–Venant equations, the lattice Boltzmann method is suitable for subcritical flows which are the most scenarios in coastal areas, estuaries and rivers. It suffer from a numerical instability when the LB method is used to solve the supercritical flows.

Reviewer#2, Concern # 3: The numerical test only include steady state solution. Does this imply the method only apply to steady state? If not, unsteady flow should be simulated.

Response: The LBCSVE method definitely can be used both in steady and unsteady flow. In the 4.1 section, the first test 'Tidal flow over a regular bed' was an unsteady flow case. The inlet water level boundary was *sine* function and changed with time. The comparison of the analytical solution and numerical water and velocity in the computational area was given in Figure 5.

Reviewer#2, Concern # 4: In the numerical tests, numerical accuracy and errors should be reported, computational efficiency compared to other numerical methods (eg. Finite difference method, finite volume method) should be provided. The effects of lattice size should also be investigated.

Response: Thanks for this valuable suggestion. Thanks for this valuable suggestion. In section 4.1, the absolute errors of water level z and velocity u are shown in Figure 5c. For the numerical models to simulate one dimensional river flows, computational time is not the restricted condition and almost every successful model can obtain the computed results. For example, for the section 4.1 Tidal flow over a regular bed, with a laptop configured by 1.8GHz CPU, 16G RAM and Intel Core i7, 21.32 seconds was consumed for the proposed LBCSVE model and 22.52 seconds was used for the Godunov-type scheme proposed by the authors^[1]. We think that the computational time and efficiency was not the point of concern when modeling the one dimensional river flows. For every test in the manuscript, the lattice size independence was carried out to find the proper lattice size.

Minor points:

Reviewer#2, Concern # 5: Figure 9, there is no need to provide area A, since it's related to surface elevation, results of flowrate should be given instead.

Response: Thanks for this valuable suggestion. We have replaced the area A by the flow rate for the Figure 9b.

Reviewer#2, Concern # 6: Figure 12, results of flowrate should be given.

Response: Thanks for this valuable suggestion. The flow rate was added in the Figure

12.

Reviewer#2, Concern # 7: There are some typos in the manuscript, for example m^2/s , superscript should be used.

Response: Thanks for this valuable suggestion. We have checked the manuscript carefully and revised the similar errors, such as '1.566 m²/s' in section 4.2 and '0.03 m/s^{1/3}' in section 4.4.

Response to Reviewer#3:

Comments to the Author(s)

The manuscript presents a lattice Boltzmann model for the Saint-Venant equations.

This topic is of potential interest but the actual contribution in the present form is still improvable. The English of the paper is good with a coherent structure. To my opinion the structure of the paper is the correct as the classic Introduction/methods/results and discussion/conclusion is followed.

The novelty of the paper is not well described and presented. To the reader (including myself) it is hard to see what is effectively the novelty of the paper since all the methodologies, as far as I could understand, have already been presented in other research. Images are of very low quality and results not perceptible in the manuscript images.

To my opinion there are some flaws in the paper that have to be clarified regarding the novelty of the paper and as such, I cannot recommend that this paper is accepted for publication in its present form. As such my advice is to review.

Comments:

- **Reviewer#3, Concern # 1:** Abstracts should not have acronyms

Response: Thanks for this valuable suggestion. The 'LBCSVE' was replaced by the 'proposed model' in the abstract.

- **Reviewer#3, Concern # 2:** P4 EQ5 - define alpha

Response: Thanks for this valuable suggestion. We have added the definition of alpha in equation 5, which is:

where α is the link in a lattice.

- **Reviewer#3, Concern # 3:** P5 L20 - the subscript should be alpha

Response: Thanks for this valuable suggestion. We have changed the Equation 20 as:

$$F_\alpha = -gA(x)S_f(x)$$

- **Reviewer#3, Concern # 4:**Eq 5 to Eq 24 and the whole methodology is similar to [18] (Van Thang, P.; Chopard, B.; Lefèvre, L. Study of the 1D lattice Boltzmann shallow water equation and its coupling to build a canal network. J. Comput. Phys. 2010, 229, 7373-7400. (doi: 10.1016/j.jcp.2010.06.022)). The only novelty I found relevant is the use of the variable A instead of h. Please state very explicitly what is the real novelty of this paper.

Response: In Thang et. al's work, the governing equations(Saint-Venant equations), are expressed as

$$\frac{\partial h}{\partial t} + \frac{\partial(hu)}{\partial x} = 0$$

$$\frac{\partial(hu)}{\partial t} + \frac{\partial(hu^2 + 0.5gh^2)}{\partial x} = F$$

The model proposed by Thang et. al is applied to simulate a canal network without considering the variation of river width and the cross-section shape. So, the model cannot be used to simulate the real rivers.

In our work, the governing equations are

$$\frac{\partial \mathbf{u}}{\partial t} + \frac{\partial \mathbf{f}}{\partial x} = \mathbf{s}$$

$$\mathbf{u} = \begin{bmatrix} A \\ Q \end{bmatrix}; \quad \mathbf{f} = \begin{bmatrix} Q \\ \frac{Q^2}{A} + gI_1 \end{bmatrix}; \quad \mathbf{s} = \begin{bmatrix} 0 \\ -gAS_f + g \frac{\partial I_1}{\partial x} \Big|_z \end{bmatrix}$$

$$I_1 = \int_0^{h_z} (h_z - h_i) b(x, h_i) dh_i$$

The primitive variables are the wetted cross-section area A and the discharge Q. The Equations are the traditional and conservative forms and can be used in real rivers with arbitrary cross-section shapes. In order to calculate the hydrostatic pressure thrust I₁ which was a difficulty in simulating the real rivers, the Gauss–Legendre numerical integration method was used.

In the introduction, page 2, we add the following statement to prove the originality for the proposed LBCSVE:

Compared with the former LB models to solve the Saint–Venant equations [19, 20],

the LBCSVE applied the conservative form of Saint–Venant equations for the first time and the Gauss–Legendre numerical integration method was used to solve the hydrostatic pressure thrust in the LB model first.

- **Reviewer#3, Concern # 5:**P6 L11 - Zhou's scheme for the force does not conserve mass locally. You should clearly state, or at least give an insight on why you have almost no errors since the scheme you used is not conservative.

Response: Using the Chapman-Enskog procedure, Zhou has verified that (1)the lattice Boltzmann equation with the centred scheme for the force term can generate the second-order accurate macroscopic continuity and momentum equations in time and space; (2) the averaged force acting on the particles during streaming can be best represented only with the centred scheme; hence the centred scheme is the correct choice for determining the force term in the lattice Boltzmann equation; (3) the centred scheme can satisfy the N-property. The four tests in the manuscript are all subcritical flows. Both the conservative form and not conservative form, very small errors is produced when simulating the subcritical flows.

- **Reviewer#3, Concern # 6:**P8 L53 - Please compute an absolute error - L2-norm or something similar is advised. Also, for each test the computational time and the machine used should be stated so that there is a comparison of efficiency between your methodology and others.

Response: Thanks for this valuable suggestion. In section 4.1, the absolute errors of water level z and velocity u are shown in Figure 5c. For the numerical models to simulate one dimensional river flows, computational time is not the restricted condition and almost every successful model can obtain the computed results. For example, for the section 4.1 Tidal flow over a regular bed, with a laptop configured by 1.8GHz CPU, 16G RAM and Intel Core i7, 21.32 seconds was consumed for the proposed LBCSVE model and 22.52 seconds was used for the Godunov-type scheme proposed by the authors^[1]. We think that the computational time and efficiency was not the point of concern when modeling the one dimensional river flows.

- **Reviewer#3, Concern # 7:**P9 L35 - Why use $v=6\text{m/s}$? please explain the value of v used for this and other tests

Response: $v = \Delta x / \Delta t$ denotes the velocity along a lattice link, with Δx being the lattice and Δt being the time step. In the lattice Boltzmann method, the v plays a key role for the numerical stability. In simulating the river flows, the stable conditions include : (1) the resultant velocity is smaller than the speed of the sound ($u_j u_j < v^2$); (2) the wave speed in shallow water is smaller than the speed of the sound ($gh < v^2$). The lattice velocity v is determined by the lattice size Δx and time step Δt . Moreover, it is must meet the stable conditions (1) and (2). For a test, the lattice velocity v varied due to the difference of lattice size Δx and time step Δt .

- **Reviewer#3, Concern # 8:** P11 L25 - m/s1/3. - fix units

Response: Thanks for this valuable suggestion. We have checked the manuscript carefully and revised the similar errors.

- **Reviewer#3, Concern # 9:** P13 Table A2 - Table is unnecessary

Response: Thanks for this valuable suggestion. The Table A2 was deleted.

- **Reviewer#3, Concern # 10:** references are formatted incorrectly. Some have an initial indentation

Response: Thanks for this valuable suggestion. We have checked the references carefully and reset the format.

References:

[1] Zhu Z, Yang Z, Bai F, et al. A New Well-Balanced Reconstruction Technique for the Numerical Simulation of Shallow Water Flows with Wet/Dry Fronts and Complex Topography[J]. Water, 2018, 10(11): 1661.